# Antigenicity and Immunogenicity Analysis of the *E. coli* Expressed FMDV Structural Proteins; VP1, VP0, VP3 of the South African Territories Type 2 Virus

**DOI:** 10.3390/v13061005

**Published:** 2021-05-27

**Authors:** Guoxiu Li, Ashenafi Kiros Wubshet, Yaozhong Ding, Qian Li, Junfei Dai, Yang Wang, Qian Hou, Jiao Chen, Bing Ma, Anna Szczotka-Bochniarz, Susan Szathmary, Yongguang Zhang, Jie Zhang

**Affiliations:** 1State Key Laboratory of Veterinary Etiological Biology, National/OIE Foot and Mouth Disease Reference Laboratory, Lanzhou Veterinary Research Institute, Chinese Academy of Agricultural Sciences, Lanzhou 730046, China; liguoxiu1122@163.com (G.L.); dingyaozhong@caas.cn (Y.D.); qianli985@163.com (Q.L.); aixinjueluofei@hotmail.com (J.D.); wangyang05@caas.cn (Y.W.); qianh910@163.com (Q.H.); m18166575233@163.com (J.C.); mabingqh@163.com (B.M.); zhangyongguang@caas.cn (Y.Z.); 2Department of Basic and Diagnostic Sciences, College of Veterinary Science, Mekelle University, Tigray 280, Ethiopia; 3Department of Swine Diseases, National Veterinary Research Institute, 57 Partyzantow, 24-100 Puławy, Poland; anna.Szczotka@piwet.pulawy.pl; 4RT-Europe Research Center, H-9200 Mosonmagyarovar, Hungary; sszathmary@gmail.com

**Keywords:** FMDV, SAT2, structural proteins, antigenicity, immunogenicity, vaccine

## Abstract

An alternative vaccine design approach and diagnostic kits are highly required against the anticipated pandemicity caused by the South African Territories type 2 (SAT2) Foot and Mouth Disease Virus (FMDV). However, the distinct antigenicity and immunogenicity of VP1, VP0, and VP3 of FMDV serotype SAT2 are poorly understood. Similarly, the particular roles of the three structural proteins in novel vaccine design and development remain unexplained. We therefore constructed VP1, VP0, and VP3 encoding gene (SAT2:JX014256 strain) separately fused with *His-SUMO* (histidine-small ubiquitin-related modifier) inserted into pET-32a cassette to express the three recombinant proteins and separately evaluated their antigenicity and immunogenicity in mice. The fusion protein was successfully expressed and purified by the Ni-NTA resin chromatography. The level of serum antibody, spleen lymphocyte proliferation, and cytokines against the three distinct recombinant proteins were analyzed. Results showed that the anti-FMDV humoral response was triggered by these proteins, and the fusion proteins did enhance the splenocyte immune response in the separately immunized mice. We observed low variations among the three fusion proteins in terms of the antibody and cytokine production in mice. Hence, in this study, results demonstrated that the structural proteins of SAT2 FMDV could be used for the development of immunodiagnostic kits and subunit vaccine designs.

## 1. Introduction

The single-stranded positive-sense RNA genome of Foot and Mouth Disease Virus (FMDV) [1] contains a large single open reading frame (ORF) post translated into structural polyproteins (VP1, VP2, VP3, and VP4) and non-structural proteins (Lab, Lb, 2A, 2B, 2C, 3A, 3B, 3C, and 3D) [2]. The structural proteins of FMDV are responsible for the assembly of viral capsids, maintenance of viral stability, binding of cells, determination of antigen specificity, and play a key role in viral infection and recognition [3]. Except for VP4, which forms the inner part of the capsid, VP1, VP2, and VP3 are partially exposed on the capsid surface [3]. The fundamental structures of VP1, VP2, and VP3 are grossly similar [4]. However, the conformation of VP1, VP2, and VP3 in different FMDV serotypes varied in the loop regions and C-terminus [5,6,7]. Several reports indicated that residues 141–160 on the GH-ring of the VP1 protein are considered as a linear B cell epitope and trigger the production of neutralizing antibody [8,9,10,11,12].

The residues 171–181 of the VP3 GH loop form folded conformations in natural particles [13] and the residues 219–221 form conformational epitope and are important for maintaining capsid stability. Additionally, the 127–140 residues of VP3 showed a strong antigenicity and were identified as a conformational epitope of the immunogenic site [14]. High genetic variability in the VP1 GH loop influences the refolding of the VP3 GH loop [13]. Altogether, VP1, VP2, and VP3 are antigenic domains of the FMDV and are highly prone to the antigenic variation [15] of the virus. This may bring variations in structural conformations and antigenicity reactions in various FMDV serotypes.

Subsequently viral evolution divides FMDV into seven major serotypes: A, O, C, Asia1, South African Territories (SAT) 1, SAT2, and SAT3 [1]. To be specific, SAT FMDV is mainly widespread in Sub-Saharan Africa. SAT2 FMD is an anticipated pandemic that could seriously threaten the livestock and pig industry around the world. As a shred of evidence, FMD outbreaks in Africa from 2000 to 2010 deposited in the world organization for animal health (OIE) annual reports confirmed that SAT2 FMDV is a predominant serotype (41%) [16]. Recently, its highest records of epidemiological jumps, transpooling, and genetic diversity labeled it as a biological threat to specific serotype-free countries and becoming a global fear. Some research results showed that the recombinant structural protein constructed by recombinant DNA technology can effectively elicit the immune response against SAT2 FMDV.

Despite this, very few studies in antigenicity and immunogenicity evaluations of VP1, VP0, and VP3 of incursionary SAT2 FMDV are available. Furthermore, the impact of the three expressed structural proteins of SAT2 FMDV in vaccine design and the use for diagnostic purposes is poorly understood. Therefore, proper understanding of the antigenicity and immunogenicity of the separately constructed VP1, VP0, and VP3 of SAT2 FMDV in a His-Sumo-pET-32a panel in an experimental model with mice would help to develop safe, stable, and protective vaccines.

In this study, we successfully expressed and purified SAT2 FMDV structural proteins. The purified proteins effectively stimulated the mice to produce humoral and cellular immune responses. Therefore, understanding the antigenic and immunogenicity features of each structural protein for a rarely studied serotype (SAT2) helps to develop safe, stable, affordable, and protective alternative vaccines. More importantly, our findings present imperative implications and provide valuable inputs to the ongoing efforts to generate effective SAT2 FMDV vaccines to control the spread of this serotype.

## 2. Materials and Methods

### 2.1. Reagents, Primers, and Experimental Animals

All the chemical reagents and biological materials were obtained from international and local commercial companies at their highest analytical grade. The DNA Marker, DNA restriction enzymes, *E. coli* BL21 (DE3), *E. coli* DH5α, and T4 DNA ligase were from (TaKaRa Bio Inc, Dalian, China). His-tag antibodies and HRP-conjugated goat anti-mice IgG were acquired from (Abcam, Cambridge, UK) and (Sigma-Aldrich, St. Louis, MO, USA), respectively. All the three pairs of primers were synthesized and delivered by (Tsingke, Xi’an, China). In addition, the pET-32a expression vector was from (NOVAGEN, Inc, Madison, MI, USA). Thirty-five 6–8 week old female experimental BALB/c mice were used from the experimental animal center, Lanzhou Veterinary Research Institute (LVRI), Chinese Academy of Agricultural Sciences (CAAS), China, under the animal qualification certificate: SCXK (G) -2015-0001. Some other reagents and chemicals are mentioned somewhere else in this article in respect to their original source.

### 2.2. Construction of Recombinant Fusion Proteins

The gene sequence of VP1, VP0, and VP3 proteins of SAT2-Africa VII-Ghb12 were retrieved from the database with GenBank accession number JX014256. This strain was an emerging SAT2 serotype which caused a devastating outbreak in Egypt in 2012. The N-terminus of all sequences represented the structural protein that was separately flanked by His-SUMO (histidine-small ubiquitin-related modifier) fusion tags (HisSumo-VP1/VP0/VP3). All the three structural protein-encoding genes were amplified by a pair of primers detailed in (Table 1). The amplicons of these genes were directly inserted into the expression vector pET-32a, which revealed pET-32a-HisSumo-VP1, pET-32a-HisSumo-VP0, and pET-32a-HisSumo-VP3 constructs. Additionally, the His-SUMO alone was also directly inserted into pET-32a, expressed, and purified in order to obtain His-SUMO protein as the negative control. The codon usage was optimized for *E. coli* expression (GenScript; Piscataway, NJ, USA) to increase the efficiency of expression, translation initiation and termination.

### 2.3. Expression and Purification of the Recombinant Fusion Proteins

As described above, we used the pET-32a vector with N-terminal peptide containing the 6xHis tag and SUMO fusion protein for the expression of a recombinant protein of our interest. The newly constructed recombinant plasmid was transformed into *E.*
*coli* BL21 (DE3) for the expression of His-tagged VP1/VP0/VP3 Sumo fusion protein. Along with this, we expressed the native pET32-His-SUMO (without inserts) used it as a negative control. The Hisx6-SUMO tags were not removed from the recombinant FMDV proteins.

An aliquot of 1 mL of overnight *E. coli* culture was inoculated into 200 mL of Luria-Bertani (LB) medium (Sigma-Aldrich, USA) and incubated at 37 °C for about 1.5 h. When the culture reached the mid-log phase (OD600 ≈ 0.8), the protein expression was induced by adding IPTG (Isopropyl-β-D-1-thiogalactopyranoside) at final concentration of 1 mM. Approximately 16 h after the IPTG induction at 16 °C, the cell pellets were harvested by centrifugation at 5000× *g* for 6 min at 4 ℃, and the pellets were resuspended in 20 mL buffer A. The cells were ultra-sonicated in an ice-filled jar, in three 5 s pulses at high intensity. The cell lysate was obtained by centrifuging at 10,000× *g* for 10 min at 4 °C. The fusion protein was purified by Ni-NTA affinity chromatography columns (Qiagen, Sacramento, CA, USA) according to the manufacturer’s instruction. The supernatant was transferred to Ni-NTA resin-containing tube after being pre-equilibrated with naïve buffer A, and then incubated for 4 h at 4 ℃. The tubes were slowly agitated to allow resins to bind with the target proteins. The recombinant protein was washed with buffer B. During purification, the following buffers were used: Buffer A (pH = 8.0): 0.1 M NaH_2_PO_4_, 0.2 M Na_2_HPO_4_, 0.3 M NaCl, 10 mM Imidazole; Buffer B (pH = 8.0): 0.1 M NaH_2_PO_4_, 0.2 M Na_2_HPO_4_, 0.3 M NaCl, 20 mM Imidazole; Buffer C (pH = 8.0): 0.1M NaH_2_PO_4_, 0.2M Na_2_HPO_4_, 0.3 M NaCl, 250 mM Imidazole. Buffer C was used to elute the recombinant proteins from the resin.

### 2.4. SDS-PAGE and Western Blotting

Sodium dodecyl sulfate polyacrylamide gel (SDS-PAGE) and Western blotting were conducted simultaneously to detect and confirm the presence and purity of the three purified fusion proteins. Primarily, 2 μg from all the purified structural proteins were electrophoresed at 12% SDS-PAGE and protein bands were visualized by Coomassie Brilliant Blue R250 (Sigma-Aldrich, St. Louis, MI, USA). Following this step, the purified structural proteins, which were electrophoresed at 12% SDS-PAGE, were transferred to polyvinylidene fluoride (PVDF) membranes and blocked with 5% (*w*/*v*) non-fat milk for 1 h at room temperature (RT) for western blotting. After washing five times with phosphate-buffered saline-Tween solution (PBST), the membranes were incubated with anti-His-tag monoclonal antibody (diluted at 1:2000) (Abcam, Cambridge, UK) overnight at 4 °C. The membranes incubated with primary antibody was washed five times with PBST, and then incubated in rabbit anti-mouse IgG antibody (Abcam, Cambridge, UK) at dilutions of 1:5000 for 1 h at RT. The washed membranes were exposed to the ECL machine using the chemiluminescence kit (Millipore Corp., Billerica, MA, USA) to verify the recombinant proteins bands.

### 2.5. Ethical Approval

In this study, all animal experiments were carried out after receiving the official approval and according to the protocol of the Institutional Animal Ethics Committee.

### 2.6. Immunization of Experimental Animals

Thirty-five 6–8 week old Specific-pathogen-free (SPF) BALB/c female mice were randomly divided into five distinct groups (7 mice/group). The groups were: (A) PBS mock group; (B) His-SUMO negative control group; (C) His-SUMO-VP1 group; (D) His-SUMO-VP0 group; (E) His-SUMO-VP3 group (Table 2). On day 1, the mice were immunized with three fusion proteins, with His-SUMO and PBS as the negative and mock groups, respectively (Table 2). The mock group (group A) was injected twice, at 100 μL/mice with PBS only, through the intramuscular route. The negative control group for FMDV recombinant protein immunogenic studies was injected twice with the His-SUMO protein with complete and incomplete Freund’s adjuvants. On day 15, each BALB/c mouse in groups of (C, D, and E), which were previously injected intramuscularly with 10 μg of purified recombinant proteins emulsified in complete Freund’s adjuvant, were injected with booster vaccines with a similar dose as that of purified recombinant proteins mixed with incomplete Freund’s adjuvant. Serum samples were collected before injection and after the first and second injection where all mice bled at 0, 7, 14, 21, and 28 days. At 28 days, the sera were collected from the retro-orbital vein and mice in all groups were sacrificed according to the approved ethical protocol. The sera were collected and kept at −70 ℃ for further analysis. 

### 2.7. Determination of Antibody Titers by ELISA

The antibody titers against the capsid proteins injected in mice were determined by indirect-ELISA. Briefly, 96-well plates were pre-coated with 200 ng/well of SAT2 FMDV virus-like particles (VLPs) (as detailed below) and each plate was incubated at 4 °C overnight. Then the microplates were blocked with 5% skim milk (100 μL/well) at 37 °C for 2 h. After washing with phosphate-buffered saline with Tween 20 (PBST), the serum samples (diluted 1:50) were added to the microtiter wells in 100 μL and incubated for 1 h at 37 °C. Horseradish peroxidase (HRP)-conjugated goat anti-mouse secondary antibody (1:20,000) was added and incubated for 1 h at 37 °C. One TMB substrate component (Minneapolis, MN, USA) was then added to each well in 100 μL and incubated for 15 min at room temperature. The reaction was stopped by adding 100 μL/well of stop solution (1.25 M H_2_SO_4_) and the plate was read at 450nm on a microplate reader (Thermo Fisher Scientific, Waltham, MA, USA). The titer of antibody reactivity was reported as OD_450nm_ values.

The purified SAT2 FMDV VLPs which coated the 96-well ELISA plates was obtained from our laboratory (unpublished data). Briefly, the VLPs were constructed from the DNA of the SAT2-AfricaVII-Ghb12 strain with GenBank accession number (JX014256) as same as the genes of origin in the current research. pFastBac Dual vector with two strong promoters, the polyhedrin promoter and the p10 promoter, of Bac-to-Bac™ Baculovirus Expression System (Invitrogen, Carlsbad, CA, USA) was applied to effectively express VP1, VP0, and VP3 capsid proteins which automatically self-assembled into VLPs in insect cells. The VLPs were purified using sucrose density gradient centrifugation.

### 2.8. Lymphocyte Proliferation Assay

Cell Counting Kit-8 (CCK-8) (APE×BIO, Houston, TX, USA) was used to determine the proliferation ability of spleen lymphocytes of the immunized mice. We strictly followed the manufacturer’s instructions in the whole process. Briefly, at 28 dpi (day after immunization), all mice were sacrificed. The single lymphocyte suspension was prepard from the aseptically eviscerated fresh spleen and adjusted to 2.5 × 10^5^ cells/mL in RPMI-1640 complete medium containing 10% FBS and 1% penicillin/streptomycin. Splenocytes, at 150 μL/well, were cultured in 96-well plates at 37 °C with 5% CO_2_ in the presence of 5 μg/mL (50 μL) antigenic stimulus and incubated for 72 h. Then, 10 μL of CCK-8 solution was added to each well in a dark environment. After an incubation period of 4 h, the lymphocyte proliferation response was analysed by measuring the absorbance at OD_490nm_ using a microplate reader. Concanavalin A (ConA) and complete RMPI-1640 were used as the positive control and mock, respectively.

### 2.9. Analysis of Cytokine IFN-γ and IL-4 Expression Level

In this study, serum samples of all sacrificed mice (at dpi 28) were applied for the cytokine analysis. The secreted level of cytokines was measured by using commercial IFN-γ, IL-2, and IL-4 ELISA kits (NeoBioscience, Shenzhen, China), following the manufacturer’s protocol.

### 2.10. Statistical Analysis

Graph Pad Prism Inc. Prism Version 8.0 analysis software, US and Microsoft Excel were used for immunological data analysis and to produce the graphs. Accordingly, a two-sided *p* ≤ 0.05 was taken as a statistically significant variation between different variables. A one-way statistical analysis of T-test was demonstrated to figure out the mean variation in immune response among treatment groups and with the control group.

## 3. Results

In this study, the pET-32a vector fused His-SUMO VP1, VP0, and VP3 of SAT2 FMDV was expressed by BL21/DE3 competent *E-coli* and expressed proteins were obtained through optimization of the expression temperature, time, and IPTG concentrations. After purification through the nickel column affinity chromatography for VP1 and VP3, and urea for VP0, the presence of the three proteins in the expression lysate was analyzed by sodium dodecyl sulfate-polyacrylamide gel electrophoresis (SDS-PAGE) and western blot. Meanwhile, the purified proteins were injected intramuscularly in the left hind lime to immunize Balb/C mice. The characteristics of serum antibody, spleen lymphocyte proliferation, and cytokine levels were evaluated. The result of the western blots with mouse anti-his tag monoclonal antibody demonstrated the His-tag in the three fused recombinant proteins. Antibody titers in serum of mice groups injected with purified His-SUMO FMDV proteins (C, D, E) were significantly higher than those in the His-SUMO and PBS groups (*p* < 0.05). Moreover, the mouse spleen lymphocytes proliferation response and the levels of IFN-γ and IL-2 in treatment groups were also significantly higher than the negative control and mock group (*p* < 0.05).

### 3.1. Construction of Recombinant Fusion Proteins

The amplified target genes (Figure 1) were inserted into the expression vector pET-32a and designed as pET-32a- (His-SUMO-VP1, His-SUMO-VP0, and His-SUMO-VP3) recombinant plasmid constructs. The presence of PCR quantified particular target inserts (gene) in the pET-32a were verified by sequencing. To further confirm the presence and the position of the insert in expression vector plasmid, we digested the constructs with restriction enzymes (Figure 1) and the result showed that the three genes were correctly constructed.

### 3.2. Purification of Recombinant Protein and Western Blot Analysis

The engineered pET-32a was transformed into Escherichia coli BL21 (DE3) for expression by IPTG. We attempted to induce protein expression with different final concentrations of IPTG. Our results showed that the variation in IPTG concentration brought no significant change in the expression (data not shown). Furthermore, in order to achieve soluble expression, we also tried to compare the expression level at different temperatures (data not shown). We found that the optimal induction temperature for expression of the protein was 16 ℃. With this temperature, the His-SUMO--VP1 target protein and His-SUMO-VP3 target protein was expressed at the molecular weight of 63 kDa and 64 kDa, respectively, in soluble form. However, expressed His-SUMO-VP0 protein (72 kDa) was found in the inclusion body. We tried to obtain the soluble form His-SUMO-VP0 protein by changing the expression vector, but the result showed no soluble expression. Soluble fusion proteins pET-32a-His-SUMO-VP1 and pET-32a-His-SUMO-VP3 were purified by Ni-NTA resin affinity chromatography. The results showed that only the two target proteins could be conjugated to the resin. The SDS-PAGE results showed that the target protein of pET-32a-His-SUMO-VP1 was eluted at 250mmol/mL with an imidazole concentration. Purified proteins concentrated in E2-E4 for pET-32a-His-SUMO-VP1 and in lane E1-E2 for pET-32a-His-SUMO-VP3 (Figure 2A) and (Figure 2C). On the other hand, the inclusion body pET-32a-His-SUMO -VP0 was purified by being dissolved in an 8mol/L urea solution. After purification, different concentrations of urea were used for renaturation. The SDS-PAGE showed that the fusion protein of His-SUMO-VP0 was renatured successfully, and the purity of VP0 fusion proteins after renaturation could reach more than 80%. Finally, the SDS-PAGE electrophoresed purified recombinant proteins were transferred to the PVDF membrane to check purity, integrity and reactivity by Western blot analysis. Anti-his-tag monoclonal antibody detected the three fusion proteins as indicated in (Figure 3D–H). This result suggested that the three fusion proteins showed the specific anticipated reaction.

### 3.3. Detection of Specific Antibodies in Mouse Serum

In this study, antibody (IgG) responses elicited by the injection of the three fusion proteins in mice serum was determined by ELISA, as shown in Figure 4. The result showed that the sera samples from all immunized mice produced a high level of specific antibodies. During the whole immune process, we noted that the serum levels of specific antibodies in the treatment group continued to rise. All of them were significantly higher than the PBS group and the His-SUMO fusion protein group (*p <* 0.05). This data anticipated that the fusion proteins could produce effective cellular immune responses in mice. In general, vaccination with the His-SUMO-tagged capsid proteins provided a reaction to SAT2 FMDV VLP demonstrating SAT2 FMDV specificity.

### 3.4. Determination of Mouse Lymphocyte Proliferation

In vitro lymphocyte proliferative responses against the purified proteins in all groups of immunized mice were analyzed at 28 dpv by the application of the Cell Counting Kit-8 (CCK8, APE×BIO, Houston, TX, USA). The results shown in Figure 5A indicated that significantly higher lymphocyte proliferation was observed in the spleen lymphocytes of the immunized groups than the other control group (*p* < 0.01). The overall results stated that the fusion proteins could effectively stimulate the cellular immune response, which would enhance the humoral response in many ways.

### 3.5. Level of Cytokines Production

We analyzed the serum levels of cytokines (IFN-γ, IL-4 and IL-2) after 28 days of immunization to ultimately evaluate the immunogenicity elicited in mice blood. Here, as shown in Figure 5B–D, the levels of IFN-γ induced by the His-SUMO -VP1, His-SUMO -VP0 and His-SUMO-VP3 fusion protein were higher than those in the His-SUMO control group. Also, the mean production level of IL-4 shows a statistically significant variation between the treatment group and the control group (*p* < 0.05). Furthermore, the results showed that these fusion proteins could potentially orchestrate the cellular and humoral immunity against FMD virus.

## 4. Discussion

FMD is one of the most economically detrimental diseases in livestock-rich countries possibly around world, mainly due to the multiple serotypes and intratypes of the virus. Our study is focused on the incursionary South African Serotype of FMDV because this serotype was recently anticipated as a serious threat to the livestock and pig industry around the world. Moreover, the antigenicity and immunogenicity of the structural proteins of this serotype are poorly understood. We clearly understand that the application of vaccination is a key component and measurement in the control and prevention of the disease [17]. However, the present traditional vaccines have several shortcomings, including the need for expensive biological containment facilities, the escape of the virus from improper inactivation, which results in vaccinated animals becoming carriers due to residual live virus, and so on [18]. Thus, the subunit vaccine design and vaccination practices against FMDV are recently blooming as an alternative approach to solve the long-lived classical problems associated with the present vaccine.

In this study, we used E. coli expression systems as a suitable bacterial expression method, mainly because of their simplicity, rapidity, and low cost [19,20]. However, we understand that heteroproteins could not often express in large quantities, fold incorrectly, or exist only as insoluble aggregates [21]. To resolve such challenges [22], Maltose binding protein (MBP), glutathione s-transferase (GST), and small ubiquitin-like modifier (SUMO) are commonly used fusion markers [23]. For our study, the Sumo protein played an important role in obtaining large quantities of cytoplasmic expressed recombinant proteins [24]. 

Our study aimed to express the three structural fusion proteins of SAT2 FMDV in E-coli to evaluate its antigenic and immunogenic strengths by characterizing the serum antibody, the spleen lymphocyte proliferation, and the cytokine levels against soluble recombinant proteins. Accordingly, all inserts were successfully constructed into an expression vector. Similar results were reported by Nguyen H.D et al. [17,25] and Silvia L et al. [26], except for the O type of FMDV from Vietnam and for both the O1 Manisa from Turkey and the SAT 2 Zim. We were convinced that the fusion of Vp0/VP1 and Vp3 into a small ubiquitin-related modifier (SUMO) protein in *E.coli* expression facilitated efficient expression and purification [27,28]. Our results indicated that the recombinant proteins of HisSumo-VP1 and HisSumo-VP3 were expressed into soluble form and purified by Ni-NTA. However, the HisSumo-VP0 found insoluble form and was aggregated in inclusion bodies as described by Chen R. et al. [29] and was exceptionally purified by urea. Therefore, the successful expression and purification of His-SUMO framed VP0, VP1, and VP3 capsid proteins of SAT2 FMDV would be an essential step forward to form VLP particle vaccine in agreement with Chen R. et al. [29]. Liu et al. also successfully expressed the FMDV VP3 protein using the pET-32a-Sumo fashion, and the recombinant protein similar to the current study had a good immune reactivity [30].

Recently, new vaccine design approaches have been emerging towards the improvement of immunogenicity and antigenicity [23]. In line with this, we designed three expressed His-SUMO fused structural proteins of SAT2 FMDV and antibody (IgG) responses elicited in the serum of immunized mice was determined by ELISA. The ELISA results in the sera samples exhibited a high level of specific antibodies. This result shows that the structural proteins are able to determine the antigenic specificity [3]. Studies have shown that the availability of free polypeptide chains in zone A and the tap-folding barrels of zone B and aspartic acid residue R56 in surface protein of VP3 could determine the binding with polyclonal antibodies [31]. Similarly, VP3 protein is crucial in recognizing the cellular heparin sulfate (HS) co-receptor of FMDV [32]. We are convinced [33] that the expression and purification structure protein of SAT2 FMDV (capsid proteins) have a great significance in the development of subunit vaccine and diagnostic kits.

In this study, the CCK method was used to measure the proliferation of lymphocytes from necropsied mice. The results showed that the spleen lymphocytes of mice immunized with the three fusion proteins display a higher proliferation compared to the PBS group and the His-SUMO group. This result indicates that these recombinant proteins successfully induced a cellular immune response against SAT2 FMDV and might do so for the other serotypes. Activated B cells are likely to produce serotype-specific neutralizing antibody responses; however, a particular serotype could induce a cross-reactive T-cell response [34]. Furthermore, the fusion tags may play its enhancing role during the stimulation with the structural proteins.

Interferons are an essential antiviral protein and are produced as a result of stronger cell-mediated immune responses in the fight against viruses like an FMD virus infection [35]. IFN-γ is one of the interferon types produced by the T cells and NK cells of the immune system. IL-2 and IL-4 are also involved in orchestrating the signaling pathway in virus recognition by cellular immunity and promote the proliferation of B cells. Therefore, our study demonstrated that the serum levels of IFN-γ induced by the HisSumo-VP1/HisSumo-VP0/HisSumo-VP3 fusion protein were higher than the His-SUMO control group. However, IL-2 and IL-4 were relatively secreted at low levels.

In this study, IFN-γ was mainly secreted by the three fusion proteins, while IL-2 and IL-4 were secreted at low levels. It has recently been reported that FMDV VP3 inhibits the expression of type I and type II interferons by reducing the mRNA levels of the virus-induced signal adapter (VISA) and degrading the JAK1 protein [2,34]. Likewise, Li et al. [18] reported that VP3 could inhibit IFN-γ induced responses and disrupts the assembly of the JAK1 complex and degrades JAK1 via a lysosomal pathway. Generally, our results confirm that the three fusion proteins could induce the secretion of certain cytokines in mice, such as IFN-γ, IL-2, and IL-4, which could promote an effective cellular immunity against the SAT2 FMDV virus.

## 5. Conclusions

In this study, the three purified fusion proteins were used to immunize mice separately. The results showed that the three fusion proteins had good antigenicity and immunogenicity compared to the control groups, which could induce effective humoral and cellular immune responses in mice. This study presented two solubly expressed fusion recombinant proteins and the VP0 in inclusion form originally from SAT2 FMD. We suggested that these proteins could be used as suitable candidates for the development of immunodiagnostic kits and subunit vaccine designs for SAT2 FMDV.

## Figures and Tables

**Figure 1 viruses-13-01005-f001:**
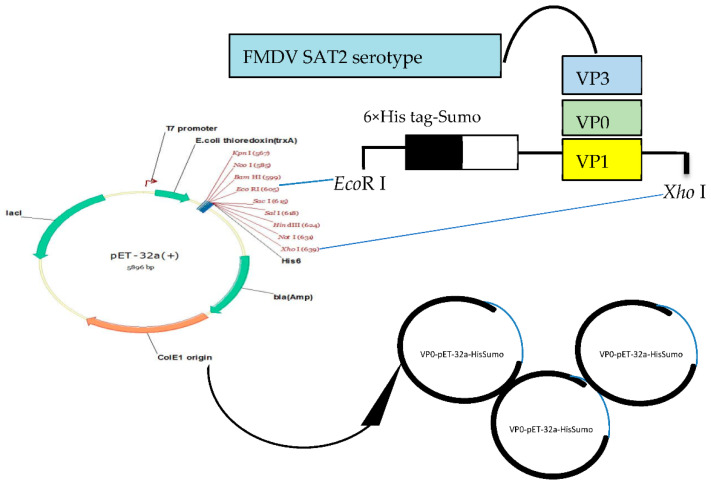
The construction of the His-SUMO fused three recombinant structural proteins of SAT2-FMDV serotype in pET-32a expression vector.

**Figure 2 viruses-13-01005-f002:**
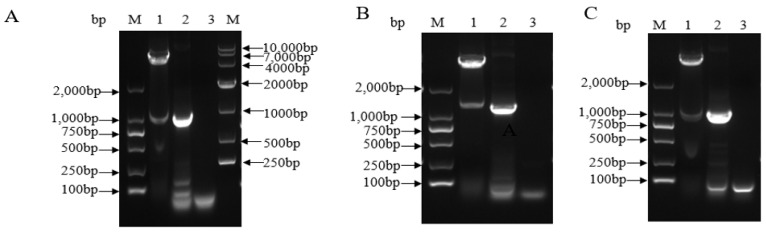
The image shows the DNA bands after Agrose gel electrophoresis; PCR amplification and restriction endonuclease digestion of His-SUMO-VP1 and VP0 in plasmid DNA. (**A**) Identification by PCR and double digestion of recombinant plasmid pET-32a- His-SUMO-VP1; lane 1: Double digestion product of pET-32a-HisSUMO-VP1; lane 2: PCR product; lane 3: Negative control; (**B**) Identification by PCR and double digestion of recombinant plasmid pET-32a- His-SUMO-VP0; lane 1: Double digestion product of pET-32a-His-SUMO-VP0; lane 2: PCR product: lane 3: Negative control; (**C**) Identification by PCR and double digestion of recombinant plasmid pET-32a- His-SUMO-VP3; lane 1: Double digestion product of pET-32a- His-SUMO -VP3; lane 2: PCR product; lane 3: Negative control.

**Figure 3 viruses-13-01005-f003:**
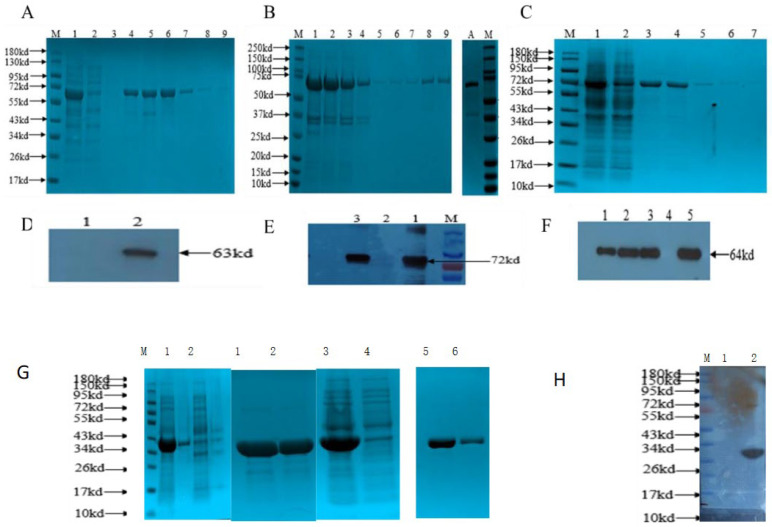
SDS-PAGE and Immunoblot analyses of the pET-32a-HisSumo-VP1/VP0/VP3 fusion protein; Fractions of the protein supernatants and antibodies and other materials were prepared as described in materials and methods. The description of the lanes in each panel is given below (**A**–**G**). (**A**) SDS-PAGE analysis of pET-32a-His-SUMO-VP1 fusion protein; Protein Molecular Weight Marker; lane 1: Supernatant of pET-32a- His-SUMO -VP1; lane 2: Flow liquid; lane 3–9: Elution liquid E1-E7. (**B**) SDS-PAGE analysis of pET-32a- His-SUMO-VP0 fusion protein; Protein Molecular Weight Marker; lane1: Inclusion body protein; lane2: Flow liquid; lane3–4: Eluted liquid E1-E2; lane5–7: Eluted liquid E3-E5; lane 8–9: Eluted liquid E6-E7; (**C**) SDS-PAGE analysis of pET-32a- His-SUMO-VP3 fusion protein. M: protein Molecular Weight Marker; 1: Supernatant of pET-32a- His-SUMO-VP1; 2: Flow liquid; 3–7: Elution liquid E1-E5. (**D**–**F**) show the result of western blot; the reactivity of the purified proteins against anti-his-tag monoclonal; (**D**) lane 1: pET-32avector; lane 2: purified pET-32a- His-SUMO-VP1 band size: 63kDa; (**E**) lane 1 and lane 3: purified pET-32a- His-SUMO-VP0 at a molecular weight of 72 kDa; lane 2: pET-32a vector; (**F**) lane 1, 2, 3, and 5: purified pET-32a- His-SUMO-VP3- at a molecular weight of 62 kDa; lane4: pET-32a vector. In conclusion, the western blot shows no reaction to the vector, only the intensity of the antigen-antibody reaction was much higher in the purified protein than in the crude. (**G**) SDS-PAGE analysis of the expressed pET-32a-His-SUMO protein found in a soluble form with the size of about 39kd; lane 1: pET-32a -His-SUMO supernatant; lane 2: pET-32a-His-SUMO sediment; lane 3: Purified pET-32a-His-SUMO protein E1; lane 4: Purified pET-32a-His-SUMO protein E2; lane 5: Purified pET-32a- His-SUMO protein E3; lane 6: Purified pET-32a- His-SUMO protein E4. (**H**) the figure shows the result of western blot; lane 1- pET32a vector only and lane 2- the purified pET-32a- His-SUMO proteins (39kd), reaction against anti-his-tag monoclonal.

**Figure 4 viruses-13-01005-f004:**
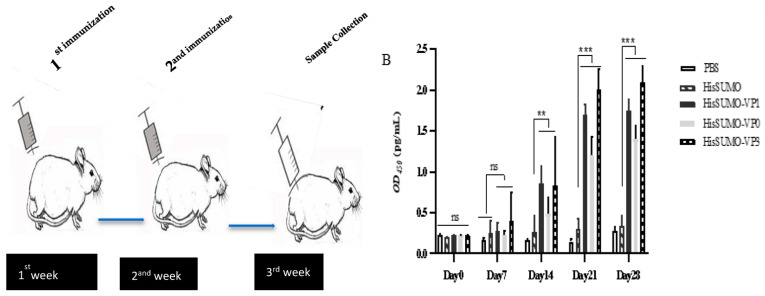
The schematic diagram shows the overall experimental design and schedule, while the graph indicates the ELISA results on the antibody responses of different experimental groups in this study. The ELISA 96-well plates were coated with FMDV-SAT2 purified VLPS and the result shown in the graph is the mean and SD of triplicate samples. The star “*”mark in the graphs denotes the presence of significant difference in each group compared to control groups (*p* < 0.05). Furthermore, the number of stars show the degree where the scientific data support significant difference among the groups, as denoted in the following: ns (no star); ** *p* < 0.01; *** *p* ≤ 0.001. Generally, the graph shows that the mean antibody titer in mice intramuscularly injected with pET-32a-HisSumo-VP1/pET-32a-HisSumo-VP0/pET-32a-HisSumo-VP3 was significantly higher (*p* < 0.01, and *p* ≤ 0.001) than that in mice injected only with PBS and pET-32a-HisSumo. (**A**) Week one and week two were the vaccination period, and blood was collected during and after vaccination. Blood collection continued on the third week and fourth week. Finally, the mice were euthanized on the fourth week. (**B**) According to the ELISA results, the titer of the anti-FMDV-VLPS antibody in mice serum before immunization and during first injection had no significant difference (*p* > 0.05) in all groups. After the booster dose, the antisera titer in the treatment groups was significantly higher than the titers in the serum of the negative control mice *p* < 0.01. The antibody response was exponentially elevated during the third and fourth weeks in groups C, D, and E and showed higher significant variation over the mock and negative control groups *p* ≤ 0.001.

**Figure 5 viruses-13-01005-f005:**
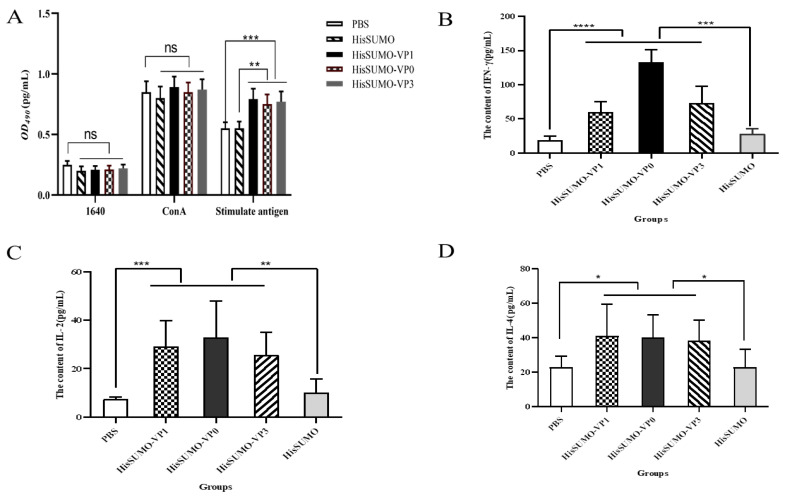
The graphs **A**–**D** show the lymphocyte proliferation and cytokine production levels elicited by the recombinant proteins, respectively. The master legend is used and each value (pg/mL) in the bar diagrams represents Mean ± S.E.M. (*n* = 7); * *p* < 0.05; ** *p* < 0.01; and *** *p* < 0.001, when compared with control groups. Graph (**A**) the absorbance at 450 nm was detected to evaluate lymphocyte proliferative concentration in (pg/mL). The results were presented as the ratio of stimulated sample to unstimulated sample at OD450 nm (stimulation index). Details of the experimental adjustments are the same as described in the methodology. Briefly, the negative control group of animals (1640) did not receive any treatment. The (ConA) which is poured by Concanavalin A (5 ug/mL), indicating the activation of B and T cells, compared with the control groups; ns = non-significant. In the group stimulated with the three recombinant FMDV proteins alone, but not treated with ConA, the concentration of lymphocytes had a significant difference ** *p* < 0.01 and *** *p* < 0.001, when compared with the control groups. Graph (**B**–**D**) The one-way ANOVA (statistical analysis of variance) was demonstrated to figure out the mean variation of the serum levels of cytokines (IFN-γ, IL-4 and IL-2) in immune response among treatment groups and control groups (ns *p* > 0.05, * *p* < 0.05, ** *p* < 0.01, *** *p* < 0.001). (**B**) The concentrations of IFN in the five different groups. The *p* values according to unpaired *t*-test **** *p* < 0.0001. (**C**) Represents the concentration of IL-2. The IL-2 concentration in mice treated with the recombinant FMDV protein compared to PBS groups **** *p* < 0.0001 and to the negative control group ** *p* < 0.01. (**D**) Shows the concentration of IL-4. The *p* values according to unpaired *t*-test * *p* < 0.05.

**Table 1 viruses-13-01005-t001:** The forward and reverse primers along with appropriate restriction endonuclease enzyme (underlined in the primer sequence) that amplify the three structural protein-encoding genes.

Primers	5′-3′ Sequences		Restriction Enzymes
VP1-F1	5′CGGAATTCCACCACCATCATCACCAC3′		EcoRI- (Forward primers) Xho I- (Reverse Primers)
VP1-R1	5′CCGCTCGAGTTACAGGGTCTGACGCTCAACG3′		
VP0-F2	5′CGGAATTCCATCATCATCATCATCACGGTAGC3′		
VP0-R2	5′CCGCTCGAGTTACTGTTTACCCGGCAGTTC3′		
VP3-F3	5′CGGAATTCCATCATCATCATCATCACGGTAGC-3′		
VP3-R3	5′CCGCTCGAGTTATTGGCGAACCGGGTCAATC-3′		
	Temperature	Time	Cycle
Initial denaturation	95 ℃	3 min	1 cycle
1st denaturation	95 ℃	30 s	
Annealing	55 ℃	30 s	34 cycles
Elongation	72 ℃	1 min	
Final elongation	72 ℃	10 min	1 cycle

**Table 2 viruses-13-01005-t002:** The five groups with respect to their immunization materials and doses of injection which were administered through the intramuscular route.

Groups	Antigen	Dose	
A	PBS	100 μL/mice	mock group
B	His-SUMO fusion protein	10 μg/mice	negative control
C	His-SUMO-VP1 fusion protein	10 μg/mice	treatment groups
D	His-SUMO-VP0 fusion protein	10 μg/mice	
E	His-SUMO-VP3 fusion protein	10 μg/mice	

## Data Availability

The datasets used and/or analyzed during the current study are obtained and available from the corresponding authors on a reasonable request.

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
