# Peer review of "Antigenicity and Immunogenicity Analysis of the E. coli Expressed FMDV Structural Proteins; VP1, VP0, VP3 of the South African Territories Type 2 Virus"

_viruses, 2021, doi:10.3390/v13061005_

Round 1

Reviewer 1 Report

In this manuscript Li et al. describe how they produced recombinant SAT2 FMDV structural proteins. They then used these proteins to immunize mice and observed immunogenic responses. Their results may constitute a useful step towards developing livestock vaccination programs to protect against FMDV infections. However, several changes in presentation should be made throughout their manuscript to enhance clarity.

Methods: In the section on immunization it is not clear if the PBS or HisSumo control groups were included in the same immunization program (administered twice, with and without adjuvant). Were mice immunized with HisSumo and adjuvant? If not, this would not be a suitable control to use as it would be unclear if mice respond to immunization with just HisSumo in the same way as the fusion proteins.

Figure 4: Text resolution is poor and the images are stretched. This makes the figure  somewhat difficult to read. Additional information should be included in the legend to  define what the stars indicate. Also, neither the legend or methods section describe the relationships of samples at different time points. Were 7 animals per group euthanized for each time point or was blood drawn from the same animals over time?

Figure 5: More information regarding conditions is required for panel A. What is "1640", "ConA", or "Stimulate antigen"? And how is the y-axis related to lymphocyte proliferation? I don't understand how the metric of pg/mL is connected to cell proliferation. All panels need to define the stars (significance thresholds) and the statistical tests that were used. Panels B-D have Chinese text in the y-axes. Is this intentional? If so, this should be described in the legends and text.

Author Response

Dear reviewer, 

We are very much pleased for your expertise comments and suggestion. We have tried our best to meet your satisfaction with our paper. Please kindly find the attached PDF file point by point response with cover letter. we are very happy to keep in touch for your upcoming comments and any suggestion on our paper to meet the publication requirements. Thank so much.

Reviewer 2 Report

In this study by Li and colleagues, the authors demonstrate that they can clone and express capsid proteins from a SAT2 serotype of foot and mouth disease virus (FMDV).  They clone separate capsid proteins (VP1, VP0 and VP3) by PCR amplification into pET32 vectors which fuse His and SUMO tags to the FMDV sequences.  The authors describe the conditions for isolation of the HisSUMO-tagged FMDV protein from the E. coli cultures and describe the purification of the proteins via Ni-NTA resin and treatment to purify the VP0 protein from inclusion bodies.  The authors use the proteins separately to immunize mice and demonstrate that the mice inoculated with the expressed FMDV capsid proteins develop antibody capable of recognizing for SAT2 FMDV VLP antigen in an ELISA test, but the mice immunized with the protein purified from the cultures of the vector without insert do not.  The authors then demonstrate that splenocytes of mice immunized in this fashion proliferate to a significantly greater extent than the mice immunized with PBS or the control protein and the splenocytes produce a significantly greater level of IFN-gamma and IL-2 and to a lesser extent, of IL-4.  The authors state that these findings provide a basis for generation of SAT2 FMDV vaccines in an economically and rapid method to control the spread of these serotypes. 

There are a lot of technical issues with this study.  A major one is the lack of evidence that the control protein called HisSUMO generated from cultures of the vector without FMDV inserts is an appropriate control.  The authors do not demonstrate anything about the control protein from expression of the plasmid vector without inserts.  In particular, is any protein purified from this system?  If not, what is being used to determine the volume of this isolation used for immunization of the mice? Specifically, if the SUMO tag is not in this control protein (which is suggested by the lack of signal using Western blots with anti-His tag monoclonal antibody to the from the control pET-32 lanes), isn’t it possible that some part of the immunization is to the SUMO tag?  The ELISA of the serum immunized uses an FMDV VLP as the antigen for the serum, which should be an indication that the immunized mice have antibody for FMDV (although the source of the FMDV SAT2 VLPs is not given nor is the method of purification of these VLPs).  However, the studies of the splenocytes does not have any comparison to FMDV antigens and could simply be measuring a response to the SUMO or His tag which are not removed in this study prior to the immunization.  This reduces the significance of this work to a demonstration that the FMDV capsid proteins can be expressed in E. coli and purified, potentially to a state which allows generation of antibodies to FMDV proteins. 

The FMDV SAT2 clone used for amplification is not noted, nor are the amplification conditions for generating the DNA used for cloning nor the orientation of the His and SUMO tags to the FMDV sequence.  The authors should refer to Yin et al. (Yin, S., Sun, S., Yang, S. et al. Self-assembly of virus-like particles of porcine circovirus type 2 capsid protein expressed from Escherichia coliVirol J 7, 166 (2010). https://doi.org/10.1186/1743-422X-7-166) to resolve at least the last issue. 

On lines 96-97, the authors state that all gene designs were optimized for E. coli expression.  In what manner were they optimized?  Does this mean a change in codon usage or just primer design?

In section 2.3, the authors refer to buffer A and B without giving any indication what the buffer is and refer to the use of Ni-NTA affinity chromatography columns according to manufacturer’s instructions without giving the manufacturer. This may seem minor but as the authors are demonstrating a method for use by others, the methods should be specified or a reference to the methods used should be given.

In legend for Figure 2, the identity of lanes 1, 2 and 3 are not given.

Figure 3 is not clearly organized.  In D, shouldn't lane 1 be the protein purified from the pET32 vector culture and lane 2, the VP1 pET32 culture? In 3E, what is lane A?

In Figure 5B, C and D, the y axis should be in English only.

From line 336 on, the numbering of the references is incorrect.

There are difficulties in English throughout.  Line 76-79, “Our results implicate and lay a foundation for possible in-vitro assembling, design and development virus like particle VLPs vaccines using thread of the three proteins.”  Not sure what thread refers to. Line 181-182, “In this study, serum sample was collected before scarification of mices (42day dpi) directly from heart champers.” probably should be “In this study, serum samples were collected directly before sacrifice of mice (at dpi 42).”.  Throughout the manuscript, mices should be mice and humeral should be humoral. 

The authors need to demonstrate that the immunization of the mice with the purified protein is not due at least to the presence of the SUMO and His tags.  One method would be to complete the purification by removing the SUMO tag with the SUMO protease or by using a control immunization with a non-FMDV protein containing a SUMO tag.  But that aside, the expression of the capsid proteins of FMDV and their use to generate VLPs capable of use as a vaccine has been done with other serotypes of FMDV and with a SAT2 FMDV.  The significance of the manuscript is not high because this is not very original. 

Author Response

(The authors gave the same response as above.)

Round 2

Reviewer 2 Report

In this study by Li and colleagues, the authors demonstrate that they can clone and express capsid proteins from a SAT2 serotype of foot and mouth disease virus (FMDV).  They clone separate capsid proteins (VP1, VP0 and VP3) by PCR amplification into pET32 vectors which fuse His and SUMO tags to the FMDV sequences.  The authors describe the conditions for isolation of the HisSUMO-tagged FMDV protein from the E. coli cultures and describe the purification of the proteins via Ni-NTA resin and treatment to purify the VP0 protein from inclusion bodies.  The authors use the proteins separately to immunize mice and demonstrate that the mice inoculated with the expressed FMDV capsid proteins develop antibody capable of recognizing for SAT2 FMDV VLP antigen in an ELISA test, but the mice immunized with the protein purified from the cultures of the vector without insert do not.  The authors then demonstrate that splenocytes of mice immunized in this fashion proliferate to a significantly greater extent than the mice immunized with PBS or the control protein and the splenocytes produce a significantly greater level of IFN-gamma and IL-2 and to a lesser extent, of IL-4.  The authors state that these findings provide a basis for generation of SAT2 FMDV vaccines in an economically and rapid method to control the spread of these serotypes. 

A major issue in my previous review was the lack of evidence that the control protein called HisSUMO generated from cultures of the vector without FMDV inserts is an appropriate control.  The authors now provide SDS-Page gels demonstrating the protein produced by the pET32-His-Sumo plasmid without FMDV insert can be isolated.  However, in Figure 3, there is no western blot using anti-His-tag monoclonal antibody of the pET32-His-Sumo protein unless in Figure 3 D, E and F lanes labeled pET32 vector, are the purified pET32-His-Sumo protein without FMDV insert.  The authors need to verify this.    

The FMDV SAT2 clone used for amplification presumably (noted in lines 210-215) is that of the Lanzhou Veterinary Research Institute.  Please provide the reference for this strain or state the history of this isolate. The amplification conditions for generating the DNA used for cloning nor the orientation of the His and SUMO tags to the FMDV sequence should also be provided. 

The ELISA of the serum immunized uses an FMDV VLP as the antigen for the serum, which is an indication that the immunized mice have antibody for FMDV.  The authors now provide a source for the FMDV VLP (lines 210-215). A statement should be inserted at lines 201 stating that the SAT2 210 FMDV VLP was generated using an insect baculovirus expression system by Lanzhou Veterinary Research Institute and referencing the production if previously published or referencing the method or providing a description of the method.

However, because the mice were immunized with the His-Sumo proteins and the His-Sumo-FMDV capsid proteins were used as stimulus for proliferation, the studies of the splenocytes does not have any comparison to native FMDV antigens and could simply be measuring a response to the SUMO or His tag which are not removed in this study prior to the immunization.  It is true that the proliferation response is significantly higher in response in mice vaccinated with the His-Sumo-FMDV capsid proteins than from mice vaccinated with the His-Sumo protein without FMDV proteins.  And this does demonstrate that a portion of the proliferative response is due to the fusion of the tags with the encoded FMDV epitopes.  If the proliferation assay had been done in response to native FMDV antigens, the anti-FMDV response would have been much more clear.  The authors state in the abstract in lines 31-32: “Results showed that the three purified fusion proteins, but not the control groups, were successfully triggered the humoral and cellular immunity against separately immunized mice. ” This is should be rewritten as the use of these proteins proves only that the anti-FMDV humoral response is triggered by these proteins and the fusion proteins do enhance a splenocyte immune response (but not specifically to FMDV). 

Lines 42-43: “Except VP4 which forms the inside part of the capsid, VP1, VP2, and VP3 completely exposed on the capsid surface [3].” VP1, VP2, and VP3 are partially exposed on the capsid surface. 

Lines 46-47: “Several reports indicated that residues 141-160 on G-H ring of VP1 protein considered as a linear B cell epitope and triggers production of neutralizing antibody” needs references.

Lines 52-53: “High genetic variability in VP1 GH loop influences the refolding of the VP3 GH loop [8]. ” This reference should be Curry S, Fry E, Blakemore W, Abu-Ghazaleh R, Jackson T, et al. (1996) Perturbations in the surface structure of A22 Iraq foot-and-mouth disease virus accompanying coupled changes in host cell specificity and antigenicity. Structure 4: 135–145.

Lines 104-105: “This is mainly ruled out that the antibodies in mice were not caused by the His-sumo fusion protein, and that the specific antibodies were produced by SAT2 FMDV VP1\VP0\VP3” The antibodies in mice included antibodies to SAT2 FMDV capsid protein as the ELISA was to a VLP of FMDV.  So, it would be better to say that vaccination with the His-Sumo-tagged capsid proteins provided a reaction to SAT2 FMDV VLP demonstrating a SAT2 FMDV specificity.

On lines 107-109, “We optimized E. coli in company (GenScript; Piscataway, NJ 08854, USA) its codon usage bias to suite to our gene designs so as to increase the efficiency of translational initiation 108 and efficiency of translational termination.” This would be clearer as, “Primer codon usage was optimized for E. coli expression (GenScript; Piscataway, NJ 08854, USA) to increase the efficiency of translation initiation and termination.” Please verify that this is correct. 

Figure 3E, what is lane A?

There still are difficulties in English.  Lines 132-134: “The newly constructed recombined plasmid was transformed into the Escherichia coli BL21 (DE3) for expression of the histidine (His)-tagged VP1/VP0/VP3 Sumo fusion protein. Along with this, we have expressed the naïve PET32-HisSUMO (without inserts) detailed below and used as a negative control.” Recombined should be recombinant and naïve should be native. Lines 237-238: “In this study, serum samples were collected directly before sacrifice of mice (at dpi 42) directly from heart champers.” Champers should be chambers.

The authors need clarify whether the protein purified from E. coli with the pET32-His-Sumo plasmid can be detected with anti-His-tag antibody as they are immunizing the control animals with this purified protein.  In addition, the authors need to discuss the fact that the purified SAT2 FMDV proteins that they are using have the SUMO tag in reference to the work on the splenocytes of immunized animals. There are numerous English usage errors that need to be corrected. But that aside, the expression of the capsid proteins of FMDV and their use to generate VLPs capable of use as a vaccine has been done with other serotypes of FMDV and with a SAT2 FMDV.  The significance of the manuscript is not high because this is not very original. 

Author Response

Dear reviewer,

Round 3

Reviewer 2 Report

In this study by Li and colleagues, the authors demonstrate that they can clone and express capsid proteins from a SAT2 serotype of foot and mouth disease virus (FMDV).  They clone separate capsid proteins (VP1, VP0 and VP3) by PCR amplification into pET32 vectors which fuse His and SUMO tags to the FMDV sequences.  The authors describe the conditions for isolation of the HisSUMO-tagged FMDV protein from the E. coli cultures and describe the purification of the proteins via Ni-NTA resin and treatment to purify the VP0 protein from inclusion bodies.  The authors use the proteins separately to immunize mice and demonstrate that the mice inoculated with the expressed FMDV capsid proteins develop antibody capable of recognizing for SAT2 FMDV VLP antigen in an ELISA test, but the mice immunized with the protein purified from the cultures of the vector without insert do not.  The authors then demonstrate that splenocytes of mice immunized in this fashion proliferate to a significantly greater extent than the mice immunized with PBS or the control protein and the splenocytes produce a significantly greater level of IFN-gamma and IL-2 and to a lesser extent, of IL-4.  The authors state that these findings provide a basis for generation of SAT2 FMDV vaccines in an economically and rapid method to control the spread of these serotypes. 

A major issue in my previous review was the lack of evidence that the control protein called HisSUMO generated from cultures of the vector without FMDV inserts is an appropriate control.  The authors now have demonstrated with a western blot using anti-His-tag monoclonal antibody that the pET32-His-Sumo protein does have the His tag and therefore likely the SUMO tag.  It is clearly from the Figure 3 H western blot that there are multiple sizes of proteins carrying the His-tag but the major band is binding the antibody strongly.  This at least provides some evidence that the immunization with the His-SUMO recombinant protein preparation provides a control in the lymphocyte studies indicating that the presence of the FMDV capsid proteins has a significant effect upon the lymphocyte response in immunized mice.  This would have been much stronger if the antigen used in the lymphocyte proliferation assay had been native FMDV antigen and not the purified recombinant proteins. 

The authors also have emphasized in their response to my review (Point 16) that the generation of a current SAT2 vaccine to deal with the prospect of the Egyptian strain spreading is important.  I note that Graham Belsham in his 2020 review of FMDV vaccines (Acta Vet Scand (2020) 62:20) stated “Vaccines need to be “matched” to the outbreak strain, not just to the serotype, to confer protection”.  I suggest the authors cite this review in the background section to highlight the need for this particular work with this strain.  In addition, the authors noted that in most studies the ability to immunized or serve as an antigen is not defined for each individual capsid protein and will provide information for future diagnostic assays and peptide vaccines.  I note that the authors have now stated this in the background section (lines 78-73). 

Although I feel the method of determining FMDV lymphocyte response could be stronger with the use of native FMDV antigen to stimulate proliferation, the authors have clarified the major issues that I had with this work.  I feel that a couple of sentences in the background will help with the clarity of the significance of the work. However there are still a large number of issues with clarity, typographic errors and English usage.

Lines 50-52: “The residues 171-181 on the VP3 GH loop encoded protein forms folded conformation in natural particle (8)”. I’m not sure what this means.  Do the authors mean: The residues 171-181 of the VP3 GH loop form folded conformations in natural particles (8)?

Lines 108-110: Vaccination with the His-Sumo-tagged capsid proteins provided a reaction to SAT2 FMDV VLP demonstrating SAT2 FMDV specificity.  This should be in the Results section 3.3 and not in the description of methods.

Lines 134-136: “Along with this, we have expressed the native PET32-HisSUMO (without inserts) detailed below and used as a negative control. But, we failed to remove from the Hisx6-SUMO tags from the proteins of our interest by protease cleavage.”

Did this fail or was this simply not done?  If it was simply not done (as I assume), please change this to:  “Along with this, we have expressed the native PET32-HisSUMO (without inserts) detailed below and used as a negative control. The Hisx6-SUMO tags have not been removed from the recombinant FMDV proteins.”

In Section 2.3, lines 139-155, was buffer C used to elute the recombinant proteins from the resin?  Please add a sentence stating this here.

Lines 231-233: “Erythrocytes artifacts removed from all plates with these cells by treating were with erythrocyte lysis buffer for 5 min at RT and twice washing with PBS.” Should be “Erythrocyte artifact was removed from all plates by treating with erythrocyte lysis buffer for 5 min at RT and twice washing with PBS”.

Lines 266-270: “Result of western blot showed that the three fused recombinant proteins were reacted to mice-anti-his-tag monoclonal antibody. Antibody titers in serum against of purified protein injected mice groups were significantly higher than those in the control groups injected with (His-sumo-protein and PBS) (p<0.05).”

I suggest: “Result of the western blots with mouse anti-his tag monoclonal antibody demonstrated the his-tag in the three fused recombinant proteins.  Antibody titers in serum of mice groups injected with purified His-SUMO FMDV proteins (D, E, F) were significantly higher than those in the control groups (His-SUMO and PBS; A and B) (p<0.05).”

Anti-His’s-Tag is used in several places and should be anti-his-tag.

Line 305: Fig2. should be Fig 3.

In the legend for Figure 3D-F, I understand that lanes labeled pET32a are a control without the recombinant FMDV proteins or the His-SUMO alone but what these proteins are is not clear. Please state in the methods section 2.3 that a control protein was produced from E. coli carrying pET32a and, if any methods were used to purify these proteins, what they were.

In Section 3.3, A, lines 390-399, this explanation of the methods should be in the methods, specifically section 2.6.  Any material not present in section 2.6 should be inserted and lines 390-399 should be deleted.

Lines 472-474: “For our study, Sumo protein has played important role to obtain large quantities cytoplasmic expressed recombinant proteins [20] and has a conserved which play role in SUMOylated mitosis [41].”  I’m not sure what  “and has a conserved which play role in SUMOylated mitosis [41]” refers to and why it is present. 

Lines 508-512: “This result indicates these recombinant proteins was successfully induced the cellular immune response against SAT2 FMDV and may be for others serotypes too. Because activated B-cells are anticipated to produce serotype-specific neutralizing antibody response; whereas a single serotype could induce a cross-reactive T-cell response [33].”

I think you mean “This result indicates these recombinant proteins successfully induced a cellular immune response against SAT2 FMDV and might do so for other serotypes too.  Activated B cells are likely to likely to produce serotype-specific neutralizing antibody responses; but a particular serotype could induce a cross-reactive T-cell response [33].”

In line 536, VP3 should be VP0.

There are spelling errors in several places:

Table 2, Negtive should be Negative and Tretament should be Treatment.

Line 230: ethunsitically scarified should be euthanized.
